# Role for Neoadjuvant Systemic Therapy for Potentially Resectable Pancreatic Cancer

**DOI:** 10.3390/cancers15082377

**Published:** 2023-04-19

**Authors:** Brandon G. Smaglo

**Affiliations:** Department of GastroIntestinal Medical Oncology, MD Anderson Cancer Center, Houston, TX 77030, USA; bgsmaglo@mdanderson.org

**Keywords:** pancreatic cancer, neoadjuvant, surgical resection

## Abstract

**Simple Summary:**

Pancreatic cancer is one of the deadliest cancers. It can only be cured by surgery, and most people with pancreatic cancer are diagnosed when the cancer is too far advanced to be able to be removed with an operation. Even when a patient can have surgery, in most instances, microscopic cancer cells will have already travelled away from the cancer and these will grow into new tumors, leading to the cancer’s eventual return. Everyone who has surgery should also get chemotherapy in an effort to prevent the cancer from recurring. This review considers if giving at least some of this chemotherapy before surgery, as opposed to giving it all after surgery, is a better approach for some patients. This may be a good idea when other aspects of the cancer make it seem to be at a higher risk for having already spread, such as a high blood test tumor marker or a very large tumor. Additionally, there are different combinations of chemotherapy drugs that could be considered for use in treating these patients. The goal of this review is to summarize which patients might be good candidates for chemotherapy before surgery and how best to treat these patients.

**Abstract:**

Despite aggressive adjuvant management, a high percentage of patients who undergo appropriate surgical resection for pancreatic cancer will see their cancer recur and thus will not be cured. An important paradigm shift to achieve better outcomes has been therapy sequence, with neoadjuvant chemotherapy preceding surgery. Patients with a borderline resectable cancer, or patients with a resectable cancer but who have other high-risk features, are ideal candidates to consider for neoadjuvant chemotherapy. Among the high-risk features, a baseline elevated CA 19-9 concentration can be particularly useful, as its response trend during neoadjuvant chemotherapy can offer important insights into the prognosis after surgery. When selecting a neoadjuvant chemotherapy regimen, response data available for the use of FOLFIRINOX and gemcitabine and nabpaclitaxel in the metastatic setting support their use in this space. FOLFIRINOX is perhaps the preferred regimen, given its proven adjuvant benefit and possibly its superior tumor response rate; still, patient tolerance and thus ability to complete recommended treatment must be carefully considered. This review presents the evidence supporting neoadjuvant chemotherapy for resectable pancreatic cancer, the factors to consider when making such a recommendation, the selection of specific regimens, and our institutional approach using these tools.

## 1. Introduction

Pancreatic cancer is on track to move up into second place in the ranking of the deadliest cancers in the United States by the end of this decade [1]. One common attribution for the high mortality rate observed with pancreas cancer is a high proportion of patients who have an advanced, inoperable disease at the time of diagnosis. However, even among those patients who have localized disease at the time of their diagnosis, five-year survival rates remain a low 42%. For comparison, the five years survival rates of a localized-at-diagnosis lung, breast, prostate, and colorectal cancer are 60%, 99%, >99%, and 91%, respectively [1]. In pancreas cancer, a high five-year mortality despite localized disease at time of diagnosis is almost always due to the development of metastatic cancer. As such, even for patients who may be candidates to potentially benefit from the surgical resection of their cancer, and thus potentially be cured of their cancer, the malignancy must still be viewed through suspicious eyes. Surgically resectable pancreas cancer should always be viewed as having a systemic component of the disease with management treating it as such.

With outcome benefits so low and treatment so complex, the approach to a patient who may potentially benefit from an operation for pancreas cancer should be both thorough and multidisciplinary. As a part of this multidisciplinary discussion, the need for adjuvant systemic therapy for essentially all patients who have surgery regardless of stage should be thought out in advance. The benefits of completing six months of adjuvant chemotherapy in conjunction with surgery for operable pancreatic cancer are well established, with the triplet chemotherapy combination FOLFIRINOX typically cited as the preferred regimen [2]. Other regimens are established in the literature with well-defined survival rates in the adjuvant setting as well as response rates in the advanced or metastatic setting.

If multidisciplinary discussion is intended to only offer surgery to those patients who may actually benefit from the operation combined with six months of toxic chemotherapy, then anything that could be undertaken to stratify a patient for a good, long-term outcome is time well spent. Some of this undertaking may be immediately available, such as the degree of tumor involvement with structures such as major blood vessels. Other aspects, however, could be learned over time. The aggressiveness of the individual tumor’s biology and its susceptibility to chemotherapy are not readily apparent in most cases upfront. However, by deferring surgery in favor of the administration of some part of the planned-for adjuvant systemic chemotherapy as a neoadjuvant treatment, the treating providers can observe the cancer’s behavioral response. This observation can consider the radiographic response of the cancer over the course of chemotherapy and can be supplemented by the observed trend in the CA 19-9 tumor marker, which is elevated to some extent in a majority of patients with pancreas cancer. This chemotherapy “stress test” can help identify the optimal time to proceed with an operation once a maximal tumor response has been observed, which in turn can be extrapolated to suggest that a similar maximal degree of systemic cancer control has been achieved. In those uncommon instances where cancer response to chemotherapy is poor, then surgical resection can be appropriately deferred, with the assumption that these are the patients most likely to have micrometastatic disease resistance to chemotherapy, which remains to eventually blossom into a cancer recurrence. Additional benefits of this neoadjuvant approach may be considered logistical, such as the easier tolerance to chemotherapy that a patient may experience by administering it when they are not in the recovery process from major surgery.

With all of this in mind, the paradigm for the treatment of patients who are candidates to potentially benefit from an operation for pancreas cancer is shifting, with many multidisciplinary teams now offering some part of the patient’s systemic chemotherapy as a neoadjuvant intervention. Requisite in this shift are considerations for which patients may benefit from such an approach, which systemic regimens to offer to patients, how best to monitor their disease response, and what benefits may be gained by taking on this risk of delaying an operation.

## 2. Role of Adjuvant Systemic Therapy

The benefit of adjuvant chemotherapy following the surgical resection of pancreatic cancer is well established. The target of adjuvant chemotherapy administration is the completion of a six-month treatment course. Due to the high risk of recurrence associated with pancreatic cancer, this recommendation is appropriate for virtually every patient who undergoes the surgical resection of pancreatic cancer, regardless of cancer stage.

The selection of a chemotherapy regimen to recommend as adjuvant should be with consideration for a patient’s likely ability to tolerate the regimen well enough to complete the entire six months. Currently, the preferred regimen for adjuvant chemotherapy is FOLFIRINOX, based upon the 2018 PRODIGE study [2]. However, this regimen is potentially quite toxic and thus not every patient is a good candidate to receive this triplet chemotherapy combination. Perhaps the best alternative to FOLFIRINOX is the doublet combination of gemcitabine and capecitabine, based upon the 2017 ESPAC-4 study [3]. FOLFIRINOX and combination gemcitabine and capecitabine have not been compared head-to-head, so the survival outcomes of each cannot be directly compared to one another. Providers should weigh expectations for tolerance against each regimens’ reported survival outcomes when making a recommendation from between these two choices.

The combination of gemcitabine and nabpaclitaxel has also been studied in the adjuvant setting. This regimen trended towards a survival advantage over a control of gemcitabine monotherapy but did not achieve its statistical endpoint [4]. As a result, in the adjuvant setting, the combination of gemcitabine and nabpaclitaxel is not a routine regimen to recommend. However, there will be individual circumstances where a fluoropyrimidine regimen shows signs of ineffectiveness as the treatment for a specific patient’s cancer. Examples might include intolerance to FOLFIRINOX that cannot be overcome or evidence of ineffectiveness if administered neoadjuvantly. In such instances, and given there was a favorable survival trend, adjuvant gemcitabine and nabpaclitaxel might be considered. Such instances should be considered as an exception and not as a standard.

Older adjuvant regimens exist that use single drug chemotherapy, specifically, single-agent 5-fluorouracil or single-agent gemcitabine [5,6]. These two regimens are established to be equivalent to one another in terms of survival outcomes, but gemcitabine emerged as the preferred option over 5-fluorouracil due to gemcitabine’s superior tolerability [7]. However, both FOLFIRINOX and the combination of gemcitabine and capecitabine were compared to single-agent gemcitabine and were established to be superior in terms of survival. As such, single-agent adjuvant chemotherapy can be considered an inferior option to the established combination options.

The survival outcomes for these various regimens are summarized in Table 1.

## 3. Adjuvant Systemic Therapy Limitations

While the benefits of adjuvant systemic therapy following the surgical resection of pancreatic cancer are well-established, this approach has room for refinement. Based upon the adjuvant data summarized, a patient who is diagnosed with pancreatic cancer that could potentially benefit from surgical resection and would undergo that operation upfront and then, once recovered from their operation, would receive a six-month course of chemotherapy. This would ideally be with FOLFIRINOX but selection for a different regimen, as summarized, might be appropriately selected as an alternative based upon the considerations for that patient’s circumstances.

In any of these approaches, survival rates remain unacceptably low, especially considering the morbidity that comes with the combined medical and surgical aspects of this care. A significant limitation of treating patients with systemic adjuvant therapy entirely in the post-operative setting is the loss of any opportunity to learn about the individual tumor’s biology in response to the adjuvant therapy administered. As the tumor will have been removed surgically from these patients, there is no longer a radiographic tumor to track for the response to chemotherapy administration. Additionally, blood biomarkers including the CA 19-9 should ideally trend down to low levels following surgical resection, and thus also are no longer available for tracking and monitoring. Therefore, in the post-operative adjuvant setting, even if the regimen is not an effective choice for an individual patient’s tumor treatment, there is no feedback available to identify this ineffectiveness, and the entire course of chemotherapy will go on to completion even though it is doomed to fail and the cancer certain to recur.

Moreover, given the high risk of recurrence with surgery for pancreas cancer, completion of the intended, recommended adjuvant course is imperative. If the patient has delays in post-operative chemotherapy due to surgical recovery or cannot complete the recommended course of chemotherapy due to their recovery or other concerns, they may be at higher risk for eventual cancer recurrence.

Therefore, any strategy that can (1) offer feedback to the treating oncologist with regard to the effectiveness of their selected chemotherapy regimen and (2) improve the likelihood of completion of the entire six-month adjuvant course would be invaluable to the patient and their long-term outcomes.

## 4. Benefits of Neoadjuvant Systemic Therapy for Pancreatic Cancer

The evidence in support of neoadjuvant chemotherapy is growing. At present, the Alliance cooperative group trial *A Phase III Trial of Perioperative Versus Adjuvant Chemotherapy for Resectable Pancreatic Cancer* is enrolling. This trial administers FOLFIRINOX perioperatively or post-operatively; with a primary outcome of overall survival, this phase III study will be pivotal to establishing the optimal sequence of treatments [8].

Pending the outcomes from this Alliance study, several studies support the benefits of neoadjuvant chemotherapy. The PREOPANC trial was a phase III study that randomly assigned patients with either a resectable or a borderline resectable cancer to treatment with either a classic post-operative gemcitabine course of adjuvant chemotherapy or to neoadjuvant, gemcitabine based, conformal radiation therapy [9]. This latter group would then undergo surgical resection, to be followed by another four months of adjuvant gemcitabine. The five-year overall survival rates were reported to be 20.5% versus 6.5% for the perioperative versus post-operative arms, respectively. While this study uses an outdated chemotherapy regimen and neoadjuvant radiation, the results do strongly support the neoadjuvant approach in concept.

Additional evidence in support of neoadjuvant systemic therapy is provided by the Japanese JSAP-05 study [10]. In this randomized Phase II/III trial, patients received either two cycles of neoadjuvant chemotherapy or were assigned to upfront surgery. In this case, the chemotherapy consisted of gemcitabine and S-1. Of note, while widely available in many parts of the world, S-1 is not currently approved for use in the United States. The media overall survival rates were 36.7 versus 26.6 months, favoring the neoadjuvant arm of the trial.

A meta-analysis published in 2019 considered 313 patients with borderline resectable pancreatic cancer from 24 studies who were treated with neoadjuvant FOLFIRINOX [11]. This analysis revealed several favorable outcomes including patient-level, median overall survival (22.2 months), resection rate (67.8%), and R0-resection rate (83.9%). These favorable results suggested further assessment in a prospective, randomized trial.

Most recently, the results of the phase II Alliance trial A021501 presented a comparison of neoadjuvant therapies. Patients had either resectable or borderline resectable cancers. All patients received neoadjuvant FOLFIRINOX chemotherapy and were randomized to receive either additional neoadjuvant radiation or not [12]. The median overall survival was reported as 29.8 months and 17.1 months for patients receiving chemotherapy alone versus chemotherapy followed by radiation, respectively, suggesting that the use of neoadjuvant FOLFIRINOX chemotherapy is an ideal reference regimen in this treatment setting. Noting that this study did not include an arm of treatment where patients were assigned to upfront surgery, the pending results of the Alliance Phase III study remain highly anticipated [8].

Given the scope of trials presently available to support neoadjuvant treatment for operable pancreas cancer, an important additional resource to consider is the consensus opinions provided by the professional societies. Guidelines from the American Society of Clinical Oncology support upfront surgery for those patients with pancreatic cancer who have a radiographically resectable tumor, but could suggest a neoadjuvant approach when the tumor is deemed borderline resectable due to tumor/vessel involvement [13]. These ASCO guidelines further support the consideration of neoadjuvant therapy in the special circumstance of a high-risk tumor, determined by an elevated baseline CA 19-9. The guidelines from the National Comprehensive Cancer Network similarly support upfront surgery for those patients who have radiographically resectable tumors and neoadjuvant considerations for those patients whose tumors are borderline resectable [14]. These guidelines further support the consideration of neoadjuvant therapy in certain high-risk groups. In addition to those patients who present with a baseline elevated CA 19-9, other such defining characteristics include patients with large or bulky tumors or nodes, or those patients presenting with high symptom burdens, such as weight loss or pain. In all cases, there are no specific thresholds for what any of these high-risk groups need to meet, and these are left as considerations for the treating physicians.

## 5. Utility of Trending the CA 19-9 in Neoadjuvant Management

If an elevated CA 19-9 baseline concentration can be considered a high-risk feature when pancreatic cancer is otherwise resectable, the trend in its concentration in response to therapy can also be an important potential indicator for benefit from neoadjuvant therapy. The ability to monitor this value serologically during treatment can offer important, real-time insights of response in a minimally invasive way that would be impractical for other regular reassessments, such as more frequent imaging. The CA 19-9 response trend can also be an important predictor for long-term outcomes after surgery.

In a retrospective evaluation, the CA 19-9 response trend was considered while patients received neoadjuvant chemotherapy who had a baseline concentration of 40 U/mL or higher and a normal total bilirubin. Those patients who had a normalization in the CA 19-9 prior to surgery had a better long-term survival than those who had a CA 19-9 that remained elevated [15]. This suggests that the use of neoadjuvant therapy when a patient has a baseline high-risk feature of an elevated CA 19-9 concentration can be administered with a potential goal of normalizing the marker in order to improve potential long-term cancer free survival. Of note, one might expect the CA 19-9 to normalize with surgery alone. Achieving a CA 19-9 normalization through surgery would suggest similar survival outcomes would be achieved in those patients who have upfront surgery to those who receive neoadjuvant chemotherapy, and this has been shown to be the case [16]. However, among those patients who had upfront surgery, 30% were found *not* to have a normalization in their CA 19-9 post-operatively, depriving them of the survival benefit. This further supports the use of neoadjuvant chemotherapy for the treatment of these high-risk patients who have an elevated CA 19-9 at baseline, where the chemotherapy-induced response in the CA 19-9 concentration can be used as a test of their cancer’s biology and thus help predict how they may benefit from surgery in the long term.

Practically speaking, the direction of the CA 19-9 over the course of neoadjuvant chemotherapy may not always be consistent. Different responses have been considered and have been shown to predict survival outcomes better than CA 19-9 normalization alone [17]. Two trends in CA 19-9 response were associated with longer-term survival:CA 19-9 that decreased from the initiation of therapy and ultimately normalized;CA 19-9 that initially rose but then reversed, decreased, and ultimately normalized.

Two trends in CA 19-9 response were associated with poor outcomes:CA 19-9 that decreased from the initiation of therapy but plateaued at a higher-than-normal concentration;CA 19-9 that rose despite therapy.

The consideration of these variations on the theme of response can help to better use the CA 19-9 trend in response to neoadjuvant chemotherapy as a pre-surgical predictor for possible outcomes.

There are important considerations that may limit the utility of CA 19-9 testing at baseline as a marker for high-risk patients. Roughly 5–10% of the population have a negative phenotype for Lewis blood group antigen expression, and thus will be so called CA 19-9 non-producers [18]. When considering patient risk based upon the CA 19-9 baseline concentration, it is important to distinguish between a low concentration and an absence of any detectable antigen at all. Additionally, a common presentation of pancreatic cancer may be with jaundice due to the tumor obstruction of the biliary system, which will cause a rise in CA 19-9 concentrations unrelated to the true extent of cancer activity [19]. This will limit the specificity of the test for these patients as it relates to their cancer burden until the obstructive jaundice has been addressed.

## 6. Neoadjuvant Chemotherapy Regimen Selection

When neoadjuvant chemotherapy is recommended, an important consideration is regimen selection. This could be determined based upon tumor response rates. Since these data are not available from the studies of purely post-operative adjuvant chemotherapy, the response rates from studies considering treatment in the metastatic setting are an important tool.

FOLFIRINOX is the one chemotherapy regimen to be established as both an optimal treatment for patients in both the front-line metastatic setting and the adjuvant setting. Its use in the metastatic setting was established by a Phase III trial from the PRODIGE group, which compared FOLFIRINOX to gemcitabine [20]. The objective response rate was 31.6% in those patients treated with FOLFIRINOX, compared to 9.4% in the control arm that used gemcitabine. Thus, FOLFIRINOX is an optimal regimen for those patients with a performance status that is adequate to tolerate this intense treatment.

In the metastatic setting, there is a proven benefit for the use of gemcitabine and nabpaclitaxel to treat pancreatic cancer. The Phase III MPACT trial comparing this doublet combination to gemcitabine monotherapy was conducted, also in the front line, metastatic setting [21]. Unlike the results of the adjuvant study using gemcitabine and nabpaclitaxel, in the metastatic setting, this doublet did prove superior to gemcitabine alone, meeting its primary endpoint of an improved overall survival. For consideration as a neoadjuvant regimen, a 23% response rate reported for gemcitabine and nabpaclitaxel can be cited, compared to 7% in the gemcitabine monotherapy control arm.

These two trials, having been conducted independently, cannot be compared to one another. Both trials demonstrated a meaningful response rate from the patients’ tumors, suggesting that, from a response perspective, either could be a good option for neoadjuvant treatment.

In attempting to compare these two regimens, a very important additional source of evidence is the NAPOLI-3 study, which at time of this manuscript’s writing, has only been presented in abstract form, at the 2023 ASCO Gastrointestinal Cancers Symposium. NAPOLI-3 was a phase 3 study comparing NALIRIFOX to gemcitabine and nabpaclitaxel in the front line, metastatic setting for patients with pancreatic cancer [22]. This was not an adjuvant or neoadjuvant study but a study in the metastatic setting, so considering its applicability in the peri-operative space is by extension. Additionally, the NALIRIFOX regimen uses liposomal irinotecan in place of conventional CPT-11, and thus is a surrogate regimen for FOLFIRINOX as discussed elsewhere in this manuscript. Nevertheless, this study has demonstrated a superior survival outcome for the treatment of patients with the triplet, 5-Fluorouracial based regimen over the gemcitabine and nabpaclitaxel doublet (median overall survival 11.1 versus 9.2 months, respectively). Interestingly, the response rates for both arms of the study were greater than what has been previously reported in other studies. Anticipating the publication of the NALIRIFOX data, extrapolating the NAPOLI-3 data into the peri-operative space may offer further support for the selection of a 5-Fluorouracial triplet regimen (FOLFIRINOX or NALIRIFOX) as the preferred neoadjuvant regimen. The data for these metastatic trials are summarized in Table 2.

The use of NALIRIFOX in the peri-operative space requires further specific study on its own. A single arm study of 168 patients treated with NALIRIFOX peri-operatively (three months neoadjuvantly and three months adjuvantly) concluded that this regimen was both manageable and active [23]. The findings support the further investigation of the NALIRIFOX regimen in the peri-operative space as well.

## 7. A Comprehensive Approach to Considering Neoadjuvant Chemotherapy for Pancreatic Cancer

The use of neoadjuvant chemotherapy for the management of patients who may be able to benefit from an operation should always be considered. Surgical resectablity based upon high quality radiographic imaging and the presence of baseline high-risk factors are critical components to this consideration. That said, neoadjuvant chemotherapy is not yet a one-size-fits-all model. For the patients who present with resectable cancer and without any high-risk features, upfront surgery remains appropriate. This consideration for treatment sequence may be made as a consensus between medical and surgical oncology, and this should occur prior to any intervention.

There is also some ambiguity as to the ideal chemotherapy regimen to offer patients in this setting, which could play a role in this recommendation. While the total amount of adjuvant therapy to be administered is established at 6 months, how much of this should be given pre-operatively when a neoadjuvant approach is employed is not fixed. Ongoing studies may further delineate the benefit of neoadjuvant therapy among certain subgroups, delineating between, for example, a patient who has a borderline resectable tumor versus a patient who has a resectable tumor at baseline.

Radiographically, if the patient’s cancer involves a major blood vessel and thus is borderline resectable, then neoadjuvant chemotherapy is indicated. When no such involvement is present, but the patient nonetheless presents with other high-risk cancer features, neoadjuvant chemotherapy is again indicated in deference to an upfront operation. These features include an elevated CA 19-9 at presentation, bulky primary cancers or lymph nodes, or advanced clinical symptoms such as significant weight loss or cancer-related pain. The response of these factors to therapy (downtrend in CA 19-9 concentration, shrinkage of tumor or nodes, improvement in clinical symptoms) can serve as important indicators for the effectiveness of this systemic therapy and thus likelihood of the patient benefiting from the surgery long term by remaining cancer free. Again, as the ongoing Alliance trial and others bear fruit, these recommendations may be further refined based on the tumor resectability subtype.

If neoadjuvant chemotherapy is recommended, two regimens should be considered: either FOLFIRINOX or gemcitabine and nabpaclitaxel. In most instances, FOLFIRINOX should be recommended to patients in the neoadjuvant setting when such a recommendation for treatment is put forward. This is for several reasons. First, a critical goal of the neoadjuvant therapy remains as a part of the 6-month, systemic adjuvant therapy intended to eliminate micrometastatic disease. FOLFIRINOX is established as perhaps the superior regimen in this regard. On the other hand, gemcitabine and nabpaclitaxel, while trending towards an adjuvant survival benefit, was not statistically superior to gemcitabine in the adjuvant setting. Second, tumor responses to FOLFIRINOX may be superior to gemcitabine and nabpaclitaxel. Both of these regimens are supported by strong response data that can be extrapolated to their use in the neoadjuvant space. To date, they have not been directly compared to one another for any response outcomes. Nevertheless, in separate studies, the response rate of cancer to FOLFIRINOX was 31.6% while that of gemcitabine and nabpaclitaxel was 23%. Finally, in the NAPOLI-3 study, the 5-Fluorouracil triplet NALIRIFOX outperformed gemcitabine and nabpaclitaxel in the metastatic space, in terms of overall survival. This is the first time a large, Phase III trial has been conducted to directly compare these two types of regimens in any line of therapy. Once response data are available from NAPOLI-3, they may be able to further clarify if a 5-Fluorouracil triplet combination is superior in terms of response when directly compared to gemcitabine and nabpaclitaxel.

This is not to say that gemcitabine and nabpaclitaxel should never be used as a neoadjuvant treatment for pancreatic cancer, as there is strong evidence in support of its effectiveness. An important inferiority of FOLFIRINOX is its significant toxicity, which can be treatment-limiting for many patients. As such, in cases where perhaps a tenuous performance status or other clinical indicators are present, the treating physician should strongly consider gemcitabine and nabpaclitaxel as a FOLFIRINOX alternative, particularly when clinical acumen indicates a higher chance of treatment receipt and thus of treatment success when the oft-better tolerated gemcitabine and nabpaclitaxel is used.

At this time, there is no strong evidence for the use of liposomal irinotecan in the peri-operative space for pancreatic cancer and thus conventional irinotecan should be used when such a drug is considered. However, this may change and evolve quickly, as further data, including response data, from the NAPOLI-3 study become available. Peri-operative trials that include liposomal irinotecan, either in the NALIRIFOX regimen or otherwise, may lead to the further evolution of this approach.

Once a neoadjuvant regimen is initiated, the same factors that supported the use of this approach over upfront surgery should be monitored for response: imaging response for borderline resectable or otherwise bulky tumors and nodes, CA 19-9 downtrend when a baseline elevated concentration was noted, improvement in clinical symptoms when these were prevalent at baseline. The CA 19-9 and clinical symptoms can be reassessed more frequently, even as often as every cycle of therapy. However, delayed responses may occur and thus an immediate improvement after one or two doses of the selected treatment regimen need not be disheartening. On the other hand, if these are worsening during the same time period, other reassessments of the disease status for chemotherapy resistance and thus progression should be hastened. In the absence of any such alarm findings, repeat imaging for response could be reasonably repeated after 6–12 weeks of therapy. If the sum interpretation of these reassessments is a tumor response, then it is reasonable to continue with ongoing neoadjuvant therapy until the effective tumor response has been seemingly maximized or unless there is a barrier to continuation, such as the accumulation of treatment-related toxicity.

Assuming a patient has received less than six months of chemotherapy neoadjuvantly, one consideration could be for the pathologic response available from the surgery when making additional recommendations for post-operative, adjuvant chemotherapy to complete that 6-month total. For tumors that have had a robust pathologic response, it is reasonable to resume the same chemotherapy regimen post-operatively to complete the remainder of the 6 months of treatment. When the response has been marginal, then longer-term outcomes for the patient could be poorer [24,25]. In this case, one might consider altering regimens to round out that 6-month total. Better studies remain necessary to more robustly support this tactic.

## 8. Future Outlooks

In all lines of therapy, the landscape for the management of pancreatic cancer continues to evolve. In the peri-operative space, there are several key areas for future study and development.

First, while the evidence is growing in support of neoadjuvant chemotherapy for many patients with pancreatic cancers who may be candidates to benefit from an operation, this approach is not settled definitively. The referenced Alliance study and other studies will continue to address this question and thus this should not be considered as settled until more mature data consistently demonstrate the benefits of this approach (testing the use). It would seem likely that there will be a subset of patients who will benefit from neoadjuvant chemotherapy and a subset who will benefit from upfront surgery, and establishing better guidelines for categorizing patients into these two categories will be key.

Second, better tools are being developed to understand the scope of a patient’s cancer, both at baseline and as it evolves in response to therapy. Recurrences that emerge after completion of surgery and chemotherapy due to resistant tumor clones but that are initially too small to appreciate on imaging or intraoperatively could be detected with better serologic tests. The CA 19-9 concentration is limited in this regard and thus is too fallible to be relied upon for significant treatment decisions such as whether or not to operate. As circulating tumor DNA and other similar technologies are refined, they will be invaluable tools to determining the true extent of a patient’s cancer and thus the optimal therapies and interventions to recommend [26].

Third, better medical therapies continue to be developed for the treatment of pancreatic cancer, and defining their roles in the peri-operative space needs to be a priority just as much as in the metastatic setting. To a surprisingly large extent, the medical management for pancreatic cancer relies on a chemotherapy-only toolkit. Hopefully, other medical options for the treatment of pancreatic cancer will prove to be effective and thus more readily available in the future. Significant efforts to define a role for immunotherapy as a treatment for pancreatic cancer continue. These include efforts to modulate the tumor microenvironment so that checkpoint inhibitors may ultimately prove effective as well as developing cellular therapies to directly target pancreatic cancers [27]. Other significant efforts continue to expand our understanding of the molecular subtypes of pancreatic cancers, and to advance our potential ability to take advantage of these findings with the development of agents that can target the key alterations in these subtypes [28]. For example, targeting KRAS alterations is an area of extensive study in the current field [29]. As these therapies evolve, they hopefully will be able to translate into the adjuvant space and thus offer a more complete adjuvant effect in those patients who are candidates to potentially benefit from surgery.

Finally, while a growing body of evidence suggests a role for the neoadjuvant application of chemotherapy in the treatment of pancreatic cancer, the role for neoadjuvant radiation therapy continues to be explored and remains more controversial. There was initially encouraging evidence for the role of neoadjuvant radiation for tumors that are deemed borderline resectable [30]. However, the previously discussed Alliance trial A021501 did not definitively support the use of neoadjuvant radiation [12]. This is still an area of exploration, especially as newer techniques for radiation delivery become refined [31,32]. In addition to medical therapy, the advancement of the understanding of the role of neoadjuvant radiation therapy may result in a truly multidisciplinary approach to the management of surgically resectable pancreatic cancer.

## 9. Conclusions

There is a growing body of evidence supporting the use of neoadjuvant chemotherapy as a treatment strategy for patients who may otherwise be candidates to benefit from an operation for pancreatic cancer. This is supported by the literature and has been adopted by major oncologic societies. Ongoing studies may further solidify support for this approach.

Neoadjuvant chemotherapy for operable pancreatic cancer is not a universal treatment recommendation, and thus each patient’s treatment strategy should undergo upfront multidisciplinary consideration. Those patients who present with a borderline resectable pancreatic cancer or who present with other high-risk features could be offered neoadjuvant chemotherapy rather than upfront surgery.

Among the high-risk features that could motivate a plan for neoadjuvant chemotherapy, a baseline CA 19-9 concentration is a particularly useful value to trend. Normalization of the concentration of this marker in response to neoadjuvant chemotherapy suggests a better long-term patient outcome and thus supports moving on to surgery.

Among the limited systemic chemotherapy options, the FOLFIRINOX regimen is preferred unless the provider feels this would be too toxic for the patient to tolerate, in which case gemcitabine and nabpaclitaxel is an acceptable alternative.

As the understanding of the nuances of pancreatic cancer’s response to therapy grows and can be more precisely predicted, the multimodality approach to potentially curable pancreatic cancer will result in ever higher long-term survival rates following surgery.

## Figures and Tables

**Table 1 cancers-15-02377-t001:** Overall survival outcomes for various adjuvant regimens.

Adjuvant Regimen	Median Overall Survival (Months)
FOLFIRINOX [2]	54.4
gemcitabine/capecitabine [3]	28
gemcitabine/nabpaclitaxel [4]	41.8
5-Fluorouracil [5]	20.1
gemcitabine [6]	22.8

**Table 2 cancers-15-02377-t002:** Outcomes of trials in the metastatic setting.

Chemotherapy Regimen	Trial	Study Groups	Objective Response Rate	Overall Survival
FOLFIRINOX	PRODIGE [20]	FOLFIRINOX vs. gemcitabine	31.6% vs. 9.4%	11.1 vs. 6.8 months
Gemcitabine + nabpaclitaxel	MPACT [21]	Gemcitabine + nabpaclitaxel vs. gemcitabine	23% vs. 7%	8.5 vs. 6.7 months
NALIRIFOX	NAPOLI-3 [22]	NALIRIFOX vs. gemcitabine + nabpaclitaxel	41.8% vs. 36.2%	11.1 vs. 9.2 months

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
