# Peer review of "Role for Neoadjuvant Systemic Therapy for Potentially Resectable Pancreatic Cancer"

_cancers, 2023, doi:10.3390/cancers15082377_

Round 1

Reviewer 1 Report

THis metanalysis should be included

J NCI 2019 111. 782-7914 Janssen et al

The text is repetitive and rambles and could be cut by at least 2 thirds to make the points better

The text would be better sequenced by development of regimens as well

Author Response

Thank you for these comments.

Concerning Janssen et al, I appreciate this recommendation and I have included this meta-analysis in section 3.

I appreciate the feedback about the text and have made some edits in an effort to address this, throughout. I did not make significant cuts to the text, based upon other feedback received from other reviewers that found the manuscript perhaps more favorable as it is. But I do appreciate the comment.

I appreciate the commentary about the order of regimens. However, in considering this point, I also endeavored to present regimens in order of some importance, even if developed in a different chronologic sequence. For example, in the adjuvant setting, I ordered the regimens based upon what one would be most to least justified in using, based upon current data.

Reviewer 2 Report

The review as it stands is quite excellent.  I might suggest adding a table summarizing the randomized trials discussed in the ms -

i.e. - Prodige, Espac, PreoPanc, Alliance 021806, Napoli 3

name, study groups, status, findings, notes

Author Response

Thank you for these comments.

I appreciate the insight of tabling the metastatic regimens. I have omitted the prior Table 2 and instead, inserted a revised Table 2 that addresses all the metastatic regimens and expands upon their outcomes.

Reviewer 3 Report

This is a review article to investigate the clinical implications of neoadjuvant systemic chemotherapy especially in patients with potentially resectable pancreatic cancer. The contents of this paper was well updated in this difficult issue. There were some issues to be clarified and revised appropriately in this manuscript as shown below.

1, Author described that “Assuming a patient hs received less than 6 months of chemotherapy neoadjuvantly, then consideration is given to the pathologic response available from surgery when making additional recommendations for post-operative ,adjuvant chemotherapy to complete that6-months total.” In page 8, line 18-21. Although this comment seems to be not generally accepted at the present time, authors should disclose background evidences suggesting this consideration.

2, The importance of further investigations for better medical therapies was described in page 8, line 46-follows. Is it better to give some comments on immune check-points inhibitors and modulators?

3,In session 4 Benefits of neoadjuvant systemic therapy for pancreatic cancer ( page 4), authors described appropriate regimen of systemic chemotherapy according to previously published literatures.

However, the reviewed papers in this session, Ref 9 and 11, were RCT studies for evaluating neoadjuvant chemotherapy regimens on the prognosis after surgery in subjected patients of both borderline resectable and resectable pancreatic cancers. The subjected patients were not only limited to resectable pancreatic cancer. Therefore, the results of these studies had a scientific value for evaluating the chemotherapy regimen not only for patients with resectable pancreatic cancer alone.

For evaluating appropriate regimens of neoadjuvant chemotherapies in patients with potentially resectable pancreatic cancer, further RCT should be awaited. Authors should clarify these points of view in the manuscript.  

4, There was nothing written about irradiation therapies in neoadjuvant therapies? I thought irradiation therapies including IMRT and heavy ion particle irradiation, might be described about its potential efficacy especially for neoadjuvant therapy in patients with resectable cancers in “Future outlooks” session.

Author Response

Thank you for these comments. I appreciate that the reviewer has provided these thoughtful and considered comments.

Concerning the first comment about the pathologic response, I have added some references and some discussion to support the poorer outcome if the response to the neoadjuvant therapy has not been robust. However, I also appreciate that the strategy of changing regimens to respond to this poorer pathological response is not well established and I have emphasized this limitation as well.

Concerning the comment about the role of other medical therapies, I appreciate this recommendation and have added some discussion of immune therapy. Along the same theme, I have additionally added some discussion about tumor genomic subtypes that may expand options for treatment.

Concerning the comment about the tumor classifications as borderline versus resectable in several of the trials, I appreciate the need to make this distinction. I have emphasized this in the section where these studies are discussed (section 3) but then also revisit this in section 6 where I revisit the ongoing Alliance trial.

Concerning the point about radiation, I appreciate this input and I have expanded the section on future outlooks to discuss the evolving and controversial role for neoadjuvant radiation in pancreatic cancer.